# Therapeutic blockade of CCL17 in obesity-exacerbated osteoarthritic pain and disease

**Kevin Ming-Chin Lee** [1]*, **Tanya Lupancu**[1], **Stacey N. Keenan**[2], **Georgina Bing**[3], **Adrian A. Achuthan**[1], **Mark Biondo**[3], **Kim Gia Lieu**[3], **Matthew J. Watt**[2], **Eugene Maraskovsky**[3], **Bronwyn A. Kingwell**[3], **John A. Hamilton**[1]

1 Faculty of Medicine Dentistry and Health Sciences, Department of Medicine, Royal Melbourne Hospital, Melbourne Medical School, University of Melbourne, Parkville, Victoria, Australia, 2 Faculty of Medicine Dentistry and Health Sciences, Department of Anatomy and Physiology, School of Biomedical Sciences, University of Melbourne, Parkville, Victoria, Australia, 3 Bio21 Institute, CSL Innovation Pty Ltd, Parkville, Victoria, Australia

* mingchinl@unimelb.edu.au

**Data Availability Statement:** All relevant data are within the manuscript and its Supporting Information files.

## Abstract

### Objectives

We previously reported that CCL17 gene-deficient mice are protected from developing pain-like behaviour and exhibit less disease in destabilization of medial meniscus (DMM)-induced OA, as well as in high-fat diet (HFD)-exacerbated DMM-induced OA. Here, we explored if therapeutic neutralization of CCL17, using increasing doses of a neutralizing monoclonal antibody (mAb), would lead to a dose-dependent benefit in these two models.

### Design

DMM-induced OA was initiated in male mice either fed with a control diet (7% fat) or 8 weeks of a 60% HFD, followed by therapeutic intraperitoneal administration (i.e. when pain is evident) of an anti-CCL17 mAb (B293, 25mg/kg, 5mg/kg or 1mg/kg) or isotype control (BM4; 25mg/kg). Pain-like behaviour and arthritis were assessed by relative static weight distribution and histology, respectively. The effects of B293 (25mg/kg) on HFD-induced metabolic changes, namely oral glucose tolerance test, insulin tolerance test and liver triglyceride levels, were examined.

### Results

Therapeutic administration of B293 results in a dramatic amelioration of DMM-induced OA pain-like behaviour and the inhibition of disease progression, compared to BM4 (isotype control) treatment. A similar therapeutic effect was observed in HFD-exacerbated OA pain-like behaviour and disease. B293 treatment did not alter the measured HFD-induced metabolic changes.

### Conclusions

Based on the data presented, CCL17 could be a therapeutic target in OA patients with joint injury alone or with obesity.

**Funding:** KMCL, TL, AAA and JAH were supported in part by funding from the University of Melbourne and CSL (Grant no. Lex34425). SNK and MJW were supported by the University of Melbourne.

**Competing interests:** KGL, MB, GB, BAK and EM are employees and shareholders of CSL. The authors have declared that no conflicts of interest exist. This does not alter our adherence to PLOS ONE policies on sharing data and materials.

## Introduction

Osteoarthritis (OA) is by far the most common musculoskeletal disease with chronic pain being the major concern for patients [1]. Treatment options for OA patients are limited and are often centered on pain relief. OA is characterized by cartilage damage, synovial inflammation and bone remodeling. There are a number of risk factors that can dramatically increase OA prevalence and severity, with obesity being prominent among these [2–4]. The OA enhancement by obesity is due not only to "so-called" mechanoinflammation (i.e. inflammation due to increased weight bearing) [5], but also to systemic low-grade inflammation (i.e. metainflammation)–several studies have implicated cytokines and adipokines in the pathogenesis of obesity-associated OA [6,7].

We have previously identified a granulocyte macrophage-colony stimulating factor (GM-CSF)/CC motif chemokine ligand 17 (CCL17) pathway in which the pro-inflammatory cytokine, GM-CSF, upregulates dramatically the production of CCL17 in monocytes/macrophages [8]. This pathway was shown to drive pain-like behaviour and disease in a number of monoarticular inflammatory mouse arthritis models [8–10]. Recently, evidence was provided, using gene-deficient mice, for the involvement of the GM-CSF/CCL17 pathway in the development of pain-like behaviour and maximal disease in three experimental OA models, as well as for the exacerbated OA development due to obesity [11]. In light of these findings, we explored the efficacy of anti-CCL17 mAb therapy for the treatment of OA in lean and obese mice.

The mouse destabilization of medial meniscus (DMM)-induced OA model is the most widely used post-traumatic OA model. Given that *Ccl17*-gene deficient mice fed either a control diet or a HFD are protected from developing DMM-induced OA pain-like behaviour and exhibit less severe arthritis [11], we reasoned it to be important to determine whether an anti-CCL17 mAb could ameliorate therapeutically DMM-induced OA pain-like behaviour and disease in a dose-dependent manner under both dietary conditions.

## Materials and methods

A detailed Materials and Methods section is available in the online S1 File.

### Anti-CCL17 (B293) monoclonal antibody and isotype control purification

The sequence for the anti-CCL17 (huB293-mG1K-aCCL17) monoclonal antibody (mAb) was derived from International PCT Publication No. WO 2015/069865 A1. Both the anti-CCL17 (B293) mAb and the isotype control (muBM4-muG1K) antibody were purified from the culture supernatant from ExpiCHO cells, which were transiently transfected with either the sequence of the anti-CCL17 mAb or the isotype control antibody. The antibodies were then sterile filtered and stored at -80˚C.

### Tango™ β-Arrestin recruitment reporter assay

Tango™ CCR4-*bla* U2OS cells (Invitrogen) were resuspended in assay medium (Thermo Fisher Scientific, FreeStyle™ 293 Expression Medium supplemented with Glutamax™) and incubated with murine CCL17 (10 nM) and increasing concentrations of B293. The fluorescence intensities at 450 nm (blue) and 520 nm (green) were used to determine the response ratio (blue: green signal) as per the manufacturer's instructions. B293 pre-complexed with murine CCL17 resulted in a dose-dependent inhibition of β-lactamase production when compared to a BM4 isotype control, indicating that B293 is able to inhibit murine CCL17-dependent signaling *in vitro*.

## Mice

Male C57BL/6 wild-type (WT) and *GM-CSF*$^{-/-}$ mice (6–10 weeks) [8,11] were used. A total of 108 mice was used in this study. All mice (n = 2–5 mice/cage) were kept under a 12-hour light and dark cycle at 22°C and allowed free access to food and water *ad libitum*. All animal experiments were approved by The University of Melbourne Animal Ethics committee (#21555).

## Destabilization of medial meniscus (DMM)-induced OA model

The mouse destabilization of medial meniscus (DMM)-induced OA model is the most widely used post-traumatic OA model. Mice were anesthetized with 4% isoflurane mixed with oxygen at a flow rate of 0.5 L/min, followed by 2% isoflurane for maintenance. DMM-induced OA was then initiated by cutting the ligament that attaches the medial meniscus to the tibia [11] in the left knee. Buprenorphine (0.1 mg/kg) was administered subcutaneously before the surgery and eight hours post-surgery to alleviate surgery-associated discomfort. Mice were killed by $CO_2$ asphyxiation for histologic analysis 12 weeks post DMM induction. For experiments examining the role of obesity, mice were fed with 60% high-fat diet (HFD) (SF02-006, Specialty Feeds) for 8 weeks prior to the initiation of the OA model [11]. All mice fed with control diet showed a slow steady increase in their body weight (approximately 15% weight increase by the end of the experiment), while all mice fed with 60% high fat diet had significant weight increase prior to the induction of the DMM OA model. For mAb treatments, mice received intraperitoneal (i.p.) injection twice weekly of 25 mg/kg, 5 mg/kg or 1 mg/kg of anti-CCL17 (B293) mAb or 25 mg/kg of isotype control antibody (BM4). These treatments commenced at week 9 for experiments with a control diet and week 7 for experiments with a HFD.

## Behavioural pain assessment

As an indicator of pain-like behaviour (referred to as pain throughout), a ratio between two knees (left *vs.* right) was used as a measure of static weight-bearing joint pain using an incapacitance meter (IITC Life Science Inc, USA) and expressed as percentage weight on the contralateral hindlimb. Values between 90 and 100 for the percentage (%) weight on the contralateral hindlimb are within a normal range of variation (i.e. no pain); a value below 90 indicates pain [8–13]. Mice were acclimatized to the incapacitance meter on at least three occasions prior to the commencement of the experiment. Three measurements were taken for each time point and averaged. All equipment was pre-calibrated.

## Histopathologic assessment of arthritis

For Safranin-O and Fast green stained knee sections, articular cartilage damage, osteophyte maturity and osteophyte size were scored using The Osteoarthritis Research Society International (OARSI) scoring system [11,14]. For H&E-stained knee sections, the degree of synovitis was assessed (**Supplemental Materials and Methods** in S1 File). All scores were acquired in a blinded manner by two independent investigators. All images are representative images of the means and taken from the mid-joint region.

## CCL17 ELISA

Sera were collected. Mouse CCL17 (R&D Systems) was measured by ELISA as per manufacturer's instructions.

### Oral glucose tolerance test and insulin tolerance test

Following 4 weeks of HFD, WT mice received intraperitoneal (i.p.). injection twice weekly of either B293 (25 mg/kg) or BM4 (25 mg/kg). WT and *GM-CSF*$^{-/-}$ mice received an oral gavage of D-glucose (2 g/kg body mass) and i.p. injection of insulin (1 U/kg body mass, Actrapid) for the oral glucose test (oGTT) and the insulin tolerance test (ITT), respectively. Blood obtained from a tail nick was assessed for glucose (Accu-Chek, Victoria, Australia) before and throughout the tests as indicated. Additional blood was obtained before and at 15 and 30 min after glucose administration. The blood was spun (2,500 g, 5 min, 4˚C) and the plasma used for later analysis of plasma insulin by ELISA (#90080, Crystal Chem, Elk Grove Village, IL).

### Quantification of liver triglyceride levels

Liver triglycerides were extracted and were determined by colorimetric assay (Triglycerides GPO-PAP; Roche Diagnostics) as previously described [15].

### Statistical analysis

For longitudinal incapacitance meter measurements, linear mixed effects models were used for repeated measures over time and a Dunnett post-hoc test was used when comparing between treatments (i.e. B293 vs. BM4) or between genotypes (WT vs. *GM-CSF*$^{-/-}$ mice). For histology measurements, the Shapiro-Wilk and Levene's test was used for assessing normality of data and homogeneity of variance. Due to the violation of normality assumption, which did not improve after the logarithmic or square root transformation, a non-parametric Kruskal-Wallis test, following Benjamini and Hochberg adjustment for *p*-values in multiple comparison, was performed to examine differences in mean histopathologic arthritis assessments, osteophyte scores and synovitis. Statistical analysis was performed using GraphPad Prism Software (10.1.2). and based on a 0.05 significance level. Plots were generated using GraphPad Prism Software (10.1.2). Data were plotted as means with corresponding 95% confidence interval (CI).

## Results

### Therapeutic efficacy of CCL17 neutralization in experimental OA

**DMM-induced OA.**   We explored the therapeutic effect of increasing doses of the anti-CCL17 (B293) mAb on the development of established DMM-induced OA pain-like behaviour and disease. The neutralizing potency of purified B293 was first assessed *in vitro* (see Materials and Methods and Supplemental Materials and Methods in S1 File). Following the initiation of DMM-induced OA [11], all WT mice developed significant pain-like behaviour by week 9 (*p = 0.004*); following sham surgery WT mice showed no pain nor disease development [11]. Following the administration of BM4 (isotype control, 25 mg/kg) (week 9), WT mice continued to exhibit pain-like behaviour until the cessation of the experiments at week 12 (**Fig 1A**). On the other hand, WT mice treated with B293 (25 mg/kg or 5 mg/kg), but not with B293 (1 mg/kg), effectively ameliorated DMM-induced OA pain-like behaviour (**Fig 1A and S1 Table**). In comparison to WT mice treated with BM4, mice treated with B293 (25 mg/kg or 5 mg/kg) also showed at termination reduced cartilage damage (**Fig 1B and 1C**) with the highest B293 dose also resulting in reduced osteophyte maturity and size (**Fig 1D and 1E**); mice treated with B293 (1 mg/kg) had comparable histologic scores to mice treated with BM4 (**Fig 1B–1E**). At termination, no differences were seen for the synovitis score between treatment groups (**Fig 1F**). Serum CCL17 concentrations were significantly elevated in mice treated

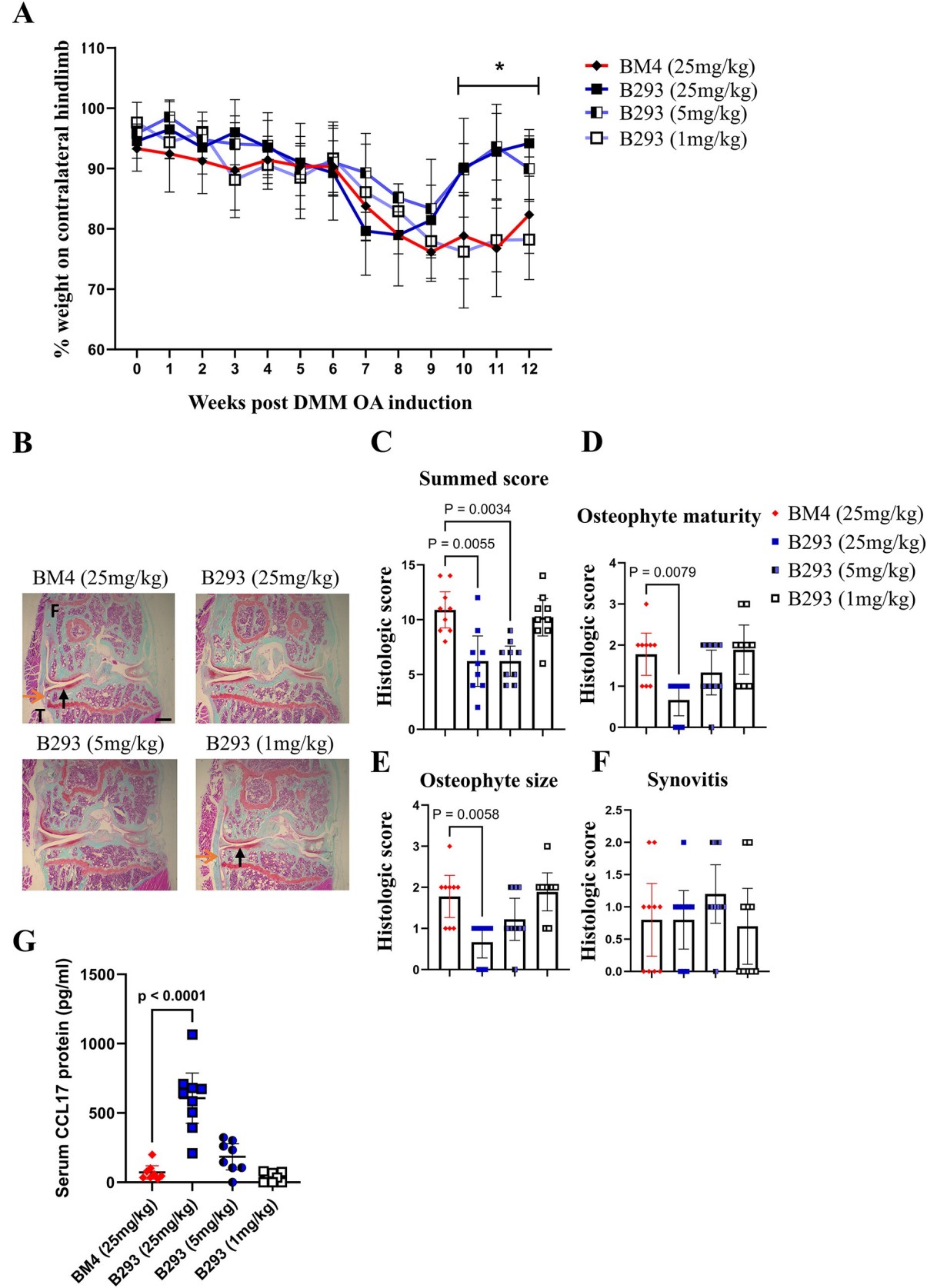

**Fig 1. Therapeutic blockade of CCL17 ameliorates DMM-induced OA pain and disease.** The DMM-induced OA model was initiated in WT male mice. The administration of B293 (anti-CCL17 mAb; 25mg/kg, 5mg/kg or 1mg/kg) or BM4 (isotype control; 25mg/kg) began at week 9 twice weekly. (A) Change in weight distribution (pain-like behaviour) over time. Numerical summaries are provided in detail in S1 Table. (B) Representative histologic pictures of knee joints (Safranin O/fast green stain, original magnification x4) for cartilage damage (indicated by black arrows) and osteophyte formation (indicated by orange arrows) (week 12). (C) DMM-induced OA cartilage damage, (D and E) osteophyte maturity and size, and (F) synovitis were quantified. (G) Sera were collected at week 12 for CCL17 quantification via ELISA. Results are expressed as the mean ± 95% CI; n = 10/treatment group; F, femur; T, tibia. Scale bar indicates 100μm. *represents p-values between BM4 vs. B293 (25mg/kg) or B293 (5mg/kg).

with B293 (25mg/kg) (**Fig 1G**), possibly due to the reported accumulation of cytokine/anti-cytokine complexes [16,17].

Thus, therapeutically administered B293 dramatically suppressed DMM OA pain-like behaviour and reduced the disease severity.

**HFD-exacerbated DMM-induced OA.** In the DMM-induced OA model, HFD-fed mice demonstrated earlier pain and more severe disease [11,18–20], which we have termed HFD-exacerbated OA [11]. We have recently provided some clinical and preclinical evidence in *Ccl17* gene deficient mice for CCL17 involvement in HFD-exacerbated OA [11]. Given the data above and obesity as a comorbidity for OA [21], we therefore tested whether therapeutic administration of B293 would be as effective in ameliorating DMM-induced OA in HFD-fed mice. Consistent with the literature [11,18–20], following the initiation of DMM-induced OA, HFD-fed WT mice developed significant pain-like behaviour by week 7 (*p = 0.0043*) (**Fig 2A**), which is earlier than for the experiment whose data are shown in **Fig 1**, and developed more severe OA disease at termination, as judged by summed histologic score (**Fig 2B and 2C**); following sham surgery HFD-fed WT mice again showed no pain nor disease development [11]. HFD-fed mice treated with B293 (25 mg/kg or 5 mg/kg) showed reversal of pain-like behaviour compared with mice treated with BM4 or B293 (1 mg/kg) (**Fig 2A and S2 Table**). HFD-fed WT mice treated with the two highest doses of B293 also showed at termination reduced cartilage damage (**Fig 2B and 2C**) and, consistent with the results in control diet-fed mice, only the highest dose of B293 reduced osteophyte maturity and size compared with the other treatment groups (**Fig 2D and 2E**). At termination, no differences were seen for the synovitis score between treatment groups (**Fig 2F**). There was again an increase in serum CCL17 levels in response to B293 administration (**Fig 2G**), which is consistent with the data shown in the experiment above (**Fig 1G**).

Together, these data demonstrate that therapeutically administered B293 also dramatically suppressed the HFD-exacerbated DMM-induced early OA pain-like behaviour and reduced the disease severity.

**CCL17 and HFD-induced metabolic changes.** CCL17 expression has been reported to be dependent on GM-CSF in a number of clinical [22–24] and preclinical studies [8–12,25,26]. More specifically, we have prior evidence, using gene deficient mice, suggesting that GM-CSF and CCL17 are both involved in obesity-associated spontaneous joint damage and the elevation of inflammatory mediators in the synovial tissue [11], which could help explain their involvement in obesity-exacerbated experimental OA. Consistent with other metabolic studies using *GM-CSF*$^{-/-}$ mice [27–30], we also observed that HFD-fed *GM-CSF*$^{-/-}$ male mice were found to be heavier than their WT counterparts (**Fig 3A**) and they showed improved glucose tolerance and had reduced plasma insulin levels during fasting and the oGTT (**Fig 3B**); insulin sensitivity, however, was similar between the genotypes as indicated by the ITT (**Fig 3C**). Additionally, liver triglyceride levels were found to be similar between the genotypes (**Fig 3D**). Based on these data, we explored the effects of anti-CCL17 mAb therapy on these obesity comorbidities.

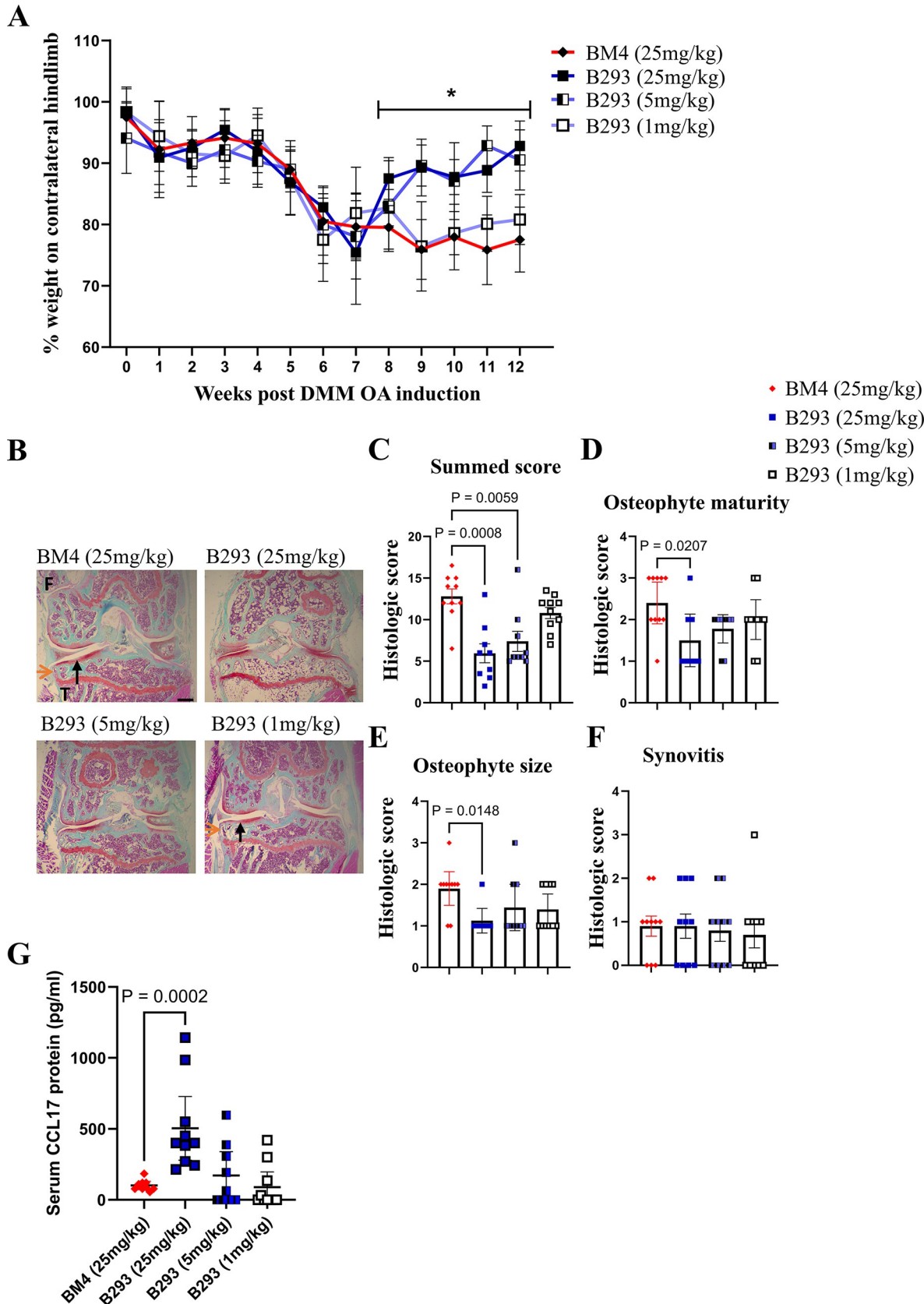

**Fig 2. Therapeutic blockade of CCL17 ameliorates HFD-exacerbated DMM-induced OA pain and disease.** WT male mice were fed with 60% high-fat diet (HFD) for 8 weeks, after which DMM-induced OA was initiated. The administration of B293 (anti-CCL17 mAb; 25mg/kg, 5mg/kg or 1mg/kg) or BM4 (isotype control; 25mg/kg) began at week 7 twice weekly. (A) Change in weight distribution (pain-like behaviour) over time. Numerical summaries are provided in detail in S2 Table. (B) Representative histologic pictures of knee joints (Safranin O/fast green stain, original magnification x4) for cartilage damage (indicated by black arrows) and osteophyte formation (indicated by orange arrows) (week 12). (C) DMM-induced OA cartilage damage, (D and E) osteophyte maturity and size, and (F) synovitis were quantified. (G) Sera were collected at week 12 for CCL17 quantification via ELISA. Results are expressed as the mean ± 95% CI; n = 10/treatment group; F, femur; T, tibia. Scale bar indicates 100μm. *represents p-values between BM4 vs. B293 (25mg/kg) or B293 (5mg/kg).

Mice were fed with a HFD for 4 weeks to induce mild obesity and glycemic dysregulation [31,32]. They were then treated with B293 (25 mg/kg) or BM4 (25 mg/kg) for a further 4 weeks, while being maintained on the HFD. Serum CCL17 levels were again increased in mice treated with B293, indicating the effectiveness of the neutralization (**Fig 4A**). Both treated groups showed a similar steady increase in body weight (**Fig 4B**), comparable responses during the oGTT and ITT (**Fig 4C and 4D**) and similar liver triglyceride content (**Fig 4E**). Together, these data suggest that CCL17 does not regulate body mass gain or obesity-associated metabolic changes in mice fed a HFD.

## Discussion

We have reported using gene-deficient male mice that CCL17 is involved in pain-like behaviour and disease in a number of OA mouse models in lean mice [11,12,25] and that this involvement is conserved in obesity-exacerbated OA models [11]. In this study, we extended these observations by exploring the therapeutic effects of targeting CCL17 using different doses of a neutralizing mAb in both DMM-induced OA and HFD-exacerbated DMM-induced OA.

The DMM surgical model in mice is a frequently used chronic OA model, with clinical relevance particularly for post-traumatic OA [33]. Following the surgical induction, mice develop moderate histologic features of OA (i.e. cartilage damage, osteophyte formation and synovitis). In the current study, we found that increasing doses of an anti-CCL17 mAb inhibit DMM-induced pain-like behaviour and reduce disease severity in male mice. Such benefit is consistent with our previous study using a single dose of a commercially available neutralizing anti-CCL17 mAb (MAB529) in a collagenase-induced OA model [12,25]. We also observed a similar therapeutic effect for HFD-exacerbated DMM-induced OA pain-like behaviour and disease in male mice. Together, these independent studies using different antibodies targeting different CCL17 epitopes and models of OA highlight the therapeutic potential of anti-CCL17 mAbs for the treatment of OA. Although the mode of action of CCL17 was not explored in the current study, CCL17-driven pain-like behaviour was previously shown to be modulated by the inhibition of neuropeptides/neurotrophins and eicosanoids [10] and *Ccl17* gene deficiency in OA synovial tissues also led to reduced gene expression of cyclooxygenase-2, an enzyme responsible for eicosanoid production [11]. One limitation of this study is that static weight bearing measurement as a pain readout does not fully recapitulate the pain phenotype observed in clinical trials. Another limitation is that this study only included male mice. It has been reported that DMM-induced OA progresses less in female mice [34]. Future studies using anti-CCL17 mAb therapy should examine other measurements of pain-like behaviour as well as examining the efficacy of anti-CCL17 mAb therapy in female mice. Interestingly, accumulation of circulating CCL17 in response to the administration of B293 was observed in both OA models tested above, consistent with other reports of increased circulating cytokine/mAb complexes following anti-cytokine mAb therapies (e.g. anti-IL-6 and anti-GM-CSF mAbs [16,17]).

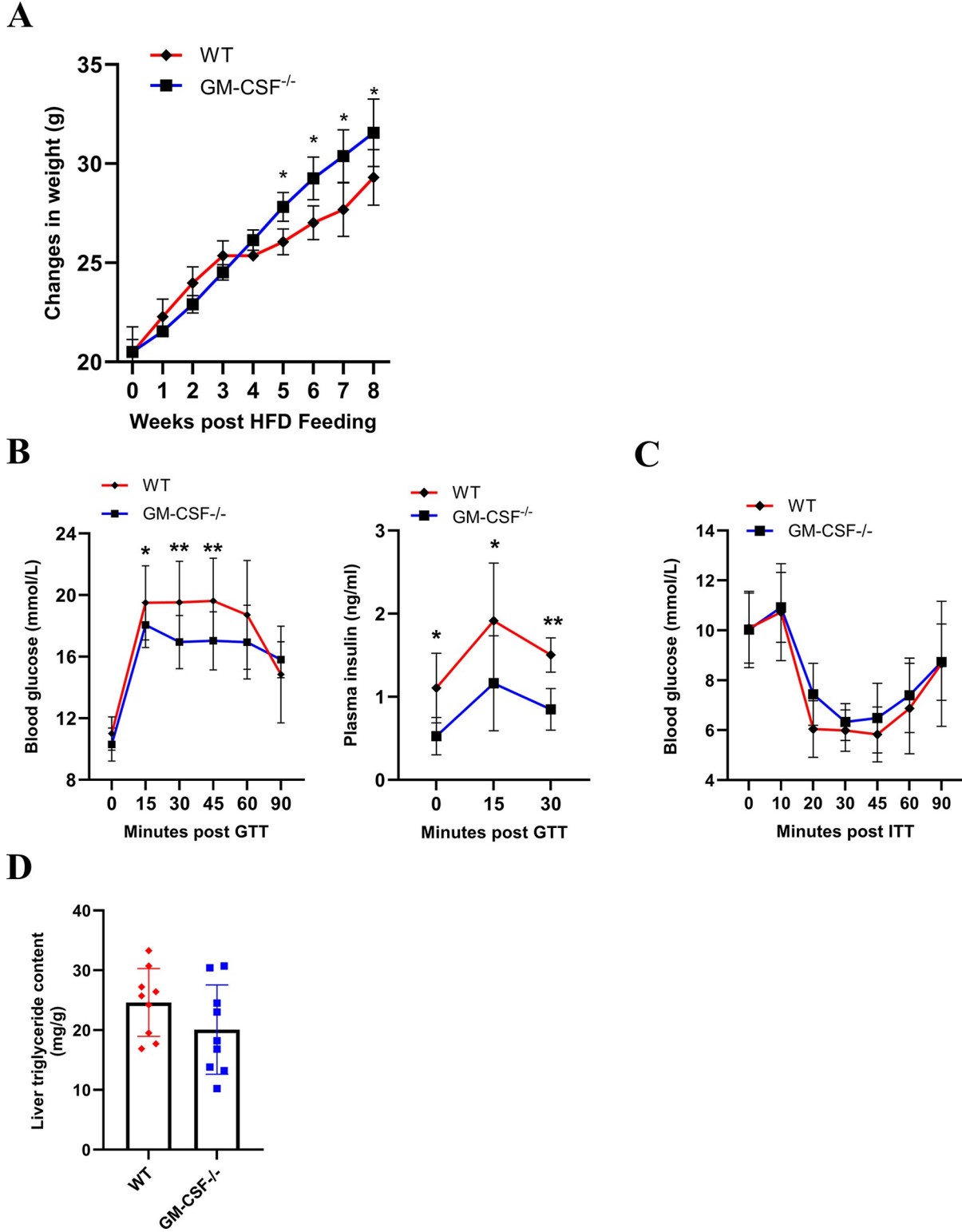

**Fig 3. The effects of GM-CSF gene deficiency on obesity-associated metabolic changes.** WT and *GM-CSF*$^{-/-}$ male mice were fed with 60% high-fat diet (HFD) for 8 weeks. (A) Absolute body weight (g). (B) Plasma glucose and insulin levels during oGTT. (C) Plasma glucose levels during ITT. (D) Triglyceride levels in liver. Results are expressed as the mean ± 95% CI; n = 9/group. *p<0.05, **p<0.01, WT vs. *GM-CSF*$^{-/-}$ mice.

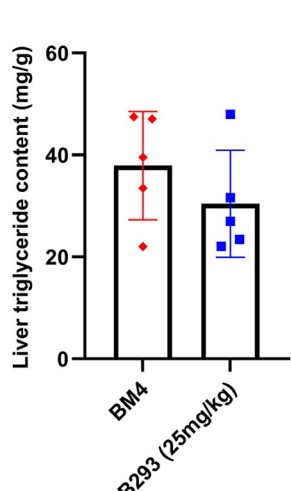

**Fig 4. The effects of CCL17 neutralization on obesity-associated metabolic changes.** WT male mice were fed with 60% high-fat diet (HFD) for 8 weeks with B293 (anti-CCL17 mAb; 25mg/kg) or BM4 (isotype control; 25mg/kg) being administered at week 4 twice weekly. (A) Sera were collected at week 8 for CCL17 quantification (ELISA). (B) Absolute body weight (g). (C) Plasma glucose and insulin levels during oGTT. (D) Plasma glucose levels during ITT. (E) Triglyceride levels in liver. (E) Sera were collected for CCL17 quantification (ELISA). Results are expressed as the mean ± 95% CI; n = 5/treatment group.

We have previously shown that mice fed with HFD have (i) elevated CCL17 levels in both sera and synovial tissue, (ii) spontaneous joint pathology and (iii) elevated cyclooxygenase-2 levels in synovial tissue, with the last two readouts being dependent on CCL17 [11]. We have also reported that obese knee OA patients have elevated circulating CCL17 levels and that synovial CCL17 gene expression correlated with body mass index [11], the circulating CCL17 levels confirming what has been reported recently in morbidly obese patients [35]. Obesity leads to the metabolic syndrome and is known to contribute to OA severity [36]. In an attempt to gain insights as to whether CCL17 could be involved in obesity-induced metabolic changes, we used the highest dose of B293 tested in this study in a model of HFD-induced obesity. While we observed some improvements in glucose tolerance in *GM-CSF*$^{-/-}$ mice (**Fig 3**), consistent with the literature [27–30], we did not see an improved glucose response in obese mice treated with B293, as judged by the oGTT and ITT (**Fig 4**)–these data are interesting given the reported dependence on CCL17 on GM-CSF actions in a number of studies [8–12,22–26]. This lack of effect of B293 treatment on obesity-induced metabolic changes could possibly be due to a sub-optimal dosing protocol for these readouts.

A recent Phase I clinical trial indicates that an anti-CCL17 mAb has shown efficacy as an analgesic in patients with knee OA pain, demonstrating favorable safety and tolerability profiles [37]. While long-term use of analgesics can be associated with radiographic progression of knee OA and an increased risk of future knee replacement [38], it is likely that anti-CCL17 mAb therapy may not have these adverse effects since our preclinical data indicate that CCL17 inhibition can provide both analgesic and cartilage protective benefits (see also references [12,39]). We suggest from our data above that CCL17 could be a therapeutic target in OA patients following joint injury alone or who are also obese for both pain and disease progression. It could also be a target in other diseases where CCL17 is expressed and where obesity is a risk factor, for example, asthma [40], and where serum levels of CCL17 are significantly elevated in the obese group [11].

## Supporting information

**S1 File. Supplemental materials and methods.**
(DOCX)

**S1 Table. Between-group mean differences in incapacitance meter analyses.**
(DOCX)

**S2 Table. Between-group mean differences in incapacitance meter analyses.**
(DOCX)

## Author Contributions

**Conceptualization:** Kevin Ming-Chin Lee, Adrian A. Achuthan, Mark Biondo, Kim Gia Lieu, Eugene Maraskovsky, Bronwyn A. Kingwell, John A. Hamilton.

**Data curation:** Kevin Ming-Chin Lee, Tanya Lupancu, Georgina Bing, Matthew J. Watt.

**Formal analysis:** Kevin Ming-Chin Lee, Tanya Lupancu, Stacey N. Keenan, Georgina Bing, Adrian A. Achuthan, Mark Biondo, Kim Gia Lieu, Matthew J. Watt, Eugene Maraskovsky, Bronwyn A. Kingwell, John A. Hamilton.

**Funding acquisition:** Kevin Ming-Chin Lee, Mark Biondo, Kim Gia Lieu, Eugene Maraskovsky, Bronwyn A. Kingwell, John A. Hamilton.

**Investigation:** Kevin Ming-Chin Lee, Tanya Lupancu, Stacey N. Keenan, Georgina Bing, Matthew J. Watt.

**Methodology:** Kevin Ming-Chin Lee, Tanya Lupancu, Stacey N. Keenan.

**Project administration:** Mark Biondo.

**Supervision:** Kevin Ming-Chin Lee, Mark Biondo, Kim Gia Lieu, Matthew J. Watt, Eugene Maraskovsky, Bronwyn A. Kingwell, John A. Hamilton.

**Validation:** Kevin Ming-Chin Lee.

**Writing – original draft:** Kevin Ming-Chin Lee, Stacey N. Keenan, Adrian A. Achuthan, Matthew J. Watt, John A. Hamilton.

**Writing – review & editing:** Kevin Ming-Chin Lee, Stacey N. Keenan, Georgina Bing, Adrian A. Achuthan, Mark Biondo, Kim Gia Lieu, Matthew J. Watt, Eugene Maraskovsky, Bronwyn A. Kingwell, John A. Hamilton.

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
