## [Decision Letter · Decision Letter 0]

11 Nov 2024

PONE-D-24-36038Therapeutic blockade of CCL17 in obesity-exacerbated osteoarthritic pain and diseasePLOS ONE

Dear Dr. Lee,

Thank you for submitting your manuscript to PLOS ONE. After careful consideration, we feel that it has merit but does not fully meet PLOS ONE’s publication criteria as it currently stands. Therefore, we invite you to submit a revised version of the manuscript that addresses the points raised during the review process.

We look forward to receiving your revised manuscript.

Kind regards,

Xindie Zhou

Academic Editor

PLOS ONE

Journal Requirements:

KGL, MB, GB, BAK and EM are employees and shareholders of CSL. The authors have declared that no conflicts of interest exist. 

5. Please provide a complete Data Availability Statement in the submission form, ensuring you include all necessary access information or a reason for why you are unable to make your data freely accessible. If your research concerns only data provided within your submission, please write "All data are in the manuscript and/or supporting information files" as your Data Availability Statement.

Additional Editor Comments :

Reviewer's comments form Reviewer 3#:

The current study aims to determine the therapeutic effects of B293 in the DMM-induced OA model. The authors have provided data on IP administration of B293 in mice and have looked at OA severity at 12-week post DMM under different conditions; however, there are many controls missing from this study. The baseline data for 9 week or 7 week for both the studies are not provided. Overall the study is premature and lacks convincing data to support the conclusion. Below are some specific comments.

The manuscript is well written but can be improved by correcting for English, for example, in lines 69-70; “mice was”; line 110 “a Dunnett post-hoc test used when”

Abstract and introduction; CCL17 gene KO mice do not develop OA or develop less severe OA?

In the materials and methods, please provide the exact time of treatment instead of “when pain like behavior was evident”

I suggest to change from “pain-like” to “pain-related” behavior.

For statistical analyses, did the author test data distribution before applying non-parametric test?

The authors have not included females in this study, which is a major limitation. There are reports which prove that female do not develop or do develop OA in DMM model.

Results:

Line 118-124: This whole paragraph is based on data not shown. the CCL13 mAb mediated inhibition is important for this study. If the authors believe that this is important to be discussed in a separate paragraph under a subheading, the authors should include the data in the manuscript instead of saying data not shown.

Line 126, authors are proposing to study the effect of B293 on the development of OA in DMM model, but administer B293 after 9 week of DMM. Multiple studies have shown that mice already develop moderate to severe OA at 8-week post DMM.

Authors have not included any data on the severity of OA at 8-week post DMM to set a baseline for the study. This time point is important to understand if B293 is inhibiting further degradation of the cartilage or the protection is through some other mechanism.

There is no sham control included in this study.

There is no no-treatment control included in this study.

The whole joint images make it difficult to see if there is any damage to the cartilage in their DMM model. The difference in the safranin o staining intensity appears be due to overstaining of safranin o as can be seen by the strong staining of growth plate which cant be seen in BM4 group image. Authors should provide better images to show cartilage damage, osteophytes.

This reviewer does not understand the rationale of combining DMM with HFD-induced OA. HFD-induced OA fits well with aging-related spontaneous OA.

Same comments for images in figure 2.

This reviewer does not understand why the results for “CCL17 and HFD-induced metabolic changes” were put in the supplementary figure.

Reviewers' comments:

Reviewer's Responses to Questions

**Comments to the Author**

1. Is the manuscript technically sound, and do the data support the conclusions?

Reviewer #1: Yes

Reviewer #2: Yes

2. Has the statistical analysis been performed appropriately and rigorously? 

Reviewer #1: Yes

Reviewer #2: Yes

3. Have the authors made all data underlying the findings in their manuscript fully available?

Reviewer #1: Yes

Reviewer #2: Yes

4. Is the manuscript presented in an intelligible fashion and written in standard English?

Reviewer #1: Yes

Reviewer #2: Yes

5. Review Comments to the Author

Reviewer #1: This is an interesting report on CCL-17 blockade in an experimental model of OA in mice (DMM°. Previous results in mice KO for CCL-17 and in a collagenase induced OA model have shown promising results.

Several points need to be clarified.

1. How many mice per group have been tested?

2. It is not made clear in DMM induced OA what was the evolution of BMI in the different groups. This is an important bias.

3. Only male mice have been selected meaning that this strategy does not work in female mice. Please comment (at least) in the discussion and explain why.

4. It does not seem that the level of pain behaviour in the first weeks is different between non fed and fed mice, while it is assume that pain may be increased in HFD-exacerbated DMM- OA ? Please comment.

5. In the histopathologic assessment where the cartilage lesions predominated?

6. Please provide information on the synovial membrane inflammation.

7. Do the authors look to anti-anti CCL-17 Ab?

General remarks

- Please comment on the risk of a long-term dramatic analgesia (according to what we have learned from the anti-NGF model) on the cartilage potential degradation.

- Please comment of the potential risk (infections) of CCL-17 inhibition.

Reviewer #2: I enjoyed reading this manuscript. Congratulations for this admirable work. These findings are very interesting indeed for relieving the OA pain in the future. I would prefer a more thorough explanation as a reader to comprehend the increased levels of CCL17 in the sera of patients with osteoarthritis despite the use of antibodies against CCl17. What is the reason that immune complexes may lead to these high levels in the sera.

6. PLOS authors have the option to publish the peer review history of their article (what does this mean?). If published, this will include your full peer review and any attached files.

Reviewer #1: No

Reviewer #2: **Yes: **Konstantina Bounia

---

## [Author Response · Author response to Decision Letter 0]

17 Dec 2024

Reviewer #1: This is an interesting report on CCL-17 blockade in an experimental model of OA in mice (DMM°. Previous results in mice KO for CCL-17 and in a collagenase induced OA model have shown promising results. Several points need to be clarified.

Response to Reviewer #1: Many thanks for your thorough review and for the opportunities for clarification. We believe we have responded appropriately to your constructive comments leading to an improved manuscript. 

1. How many mice per group have been tested?

Response to Reviewer: We already had indicated in the Figure legends that there are 10 mice per treatment group (see Figure Legends, lines 395, 407, 412 and 419). 

2. It is not made clear in DMM induced OA what was the evolution of BMI in the different groups. This is an important bias.

Response to Reviewer: All mice fed with control diet showed a slow steady increase in their body weight (approximately 15% weight increase by the end of the experiment), while all mice fed with 60% high fat diet had significant weight increase prior to the induction of the DMM OA model (Figs 3 and 4, and Materials and Methods); mice continued to increase their body weight until the end of the experiment.

3. Only male mice have been selected meaning that this strategy does not work in female mice. Please comment (at least) in the discussion and explain why.

Response to Reviewer: We have added a comment that only male mice were used in this study with a suggestion that future studies should include female mice to assess the efficacy of anti-CCL17 mAb therapy (see lines 239-243). 

4. It does not seem that the level of pain behaviour in the first weeks is different between non fed and fed mice, while it is assume that pain may be increased in HFD-exacerbated DMM- OA ? Please comment.

Response to Reviewer: We have previously reported that, in comparison to control diet-fed DMM OA mice, HFD-fed DMM OA mice develop earlier onset of pain-like behaviour (around week 9 vs. week 7, respectively) and had more severe disease (PMID 37225052). This background data provided the rationale for the current study (see lines 175-178). The pain-like behaviour between control diet-fed (Fig. 1) and high-fat diet-fed mice (Fig. 2) was similar in the first few weeks post DMM OA induction, which is consistent with our previous study (PMID 37225052). 

5. In the histopathologic assessment where the cartilage lesions predominated?

Response to Reviewer: In this study, the scoring criteria for cartilage lesions in regions of lateral tibia, latera femur, medial tibia and medial femur are in accordance with the recommendations from OARSI, which is often used in the OA DMM literature (see, for example, PMID 33609695 and PMID 29434267). 

6. Please provide information on the synovial membrane inflammation.

Response to Reviewer: We have previously reported that DMM OA wild-type mice develop some synovial membrane inflammation (i.e. synovitis) and that CCL17 KO mice have no effect on this readout (PMID 37225052). In response, we have now provided synovitis scores in the updated Figures (see new Figs. 1F and 2F). Consistently, mice treated with B293 showed no effects on synovitis compared to mice treated with BM4 (see lines 168-169 and 190-191). Relevant scoring methods are now also included in the Materials and Methods (lines 121-122) and in the Supplemental Materials and Methods (lines 51-55). 

7. Do the authors look to anti-anti CCL-17 Ab?

Response to Reviewer: It would be very interesting to know if auto-antibodies against anti-CCL17 Ab were generated. However, we did not have an assay to explore the levels of anti-anti-CCL17 Ab in the sera. 

General remarks

- Please comment on the risk of a long-term dramatic analgesia (according to what we have learned from the anti-NGF model) on the cartilage potential degradation.

Response to Reviewer: We have added comments on the risk of long-term analgesia on the potential cartilage degradation (see lines 263-267).

- Please comment of the potential risk (infections) of CCL-17 inhibition.

Response to Reviewer: We have added comments about safety profile of CCL17 inhibition (see lines 262-263).

Reviewer #2: I enjoyed reading this manuscript. Congratulations for this admirable work. These findings are very interesting indeed for relieving the OA pain in the future. I would prefer a more thorough explanation as a reader to comprehend the increased levels of CCL17 in the sera of patients with osteoarthritis despite the use of antibodies against CCl17. What is the reason that immune complexes may lead to these high levels in the sera.

Response to Reviewer #2: Many thanks for your positive comments. With regards to the elevated circulating levels of CCL17 in response to anti-CCL17 therapy, we presume that the CCL17 epitope recognized by the detection antibody from the mouse CCL17 ELISA kit (R&D Systems) is different to the binding epitope of B293; a similar approach is well described in the literature (see, for example, references 16 and 17) as a tool to monitor the effectiveness of a neutralizing anti-cytokine monoclonal antibody as a therapy. 

Reviewer's comments from Reviewer 3#:

The current study aims to determine the therapeutic effects of B293 in the DMM-induced OA model. The authors have provided data on IP administration of B293 in mice and have looked at OA severity at 12-week post DMM under different conditions; however, there are many controls missing from this study. The baseline data for 9 week or 7 week for both the studies are not provided. Overall the study is premature and lacks convincing data to support the conclusion. Below are some specific comments.

Response to Reviewer: Many thanks for your thorough review and for the opportunities for clarification. We believe we have responded appropriately to your constructive comments leading to an improved manuscript. 

The manuscript is well written but can be improved by correcting for English, for example, in lines 69-70; “mice was”; line 110 “a Dunnett post-hoc test used when”

Response to Reviewer: Many thanks for pointing it out and we have rectified these mistakes. 

Abstract and introduction; CCL17 gene KO mice do not develop OA or develop less severe OA?

Response to Reviewer: We have shown that CCL17 gene KO mice were protected from pain-like behaviour and developed less OA disease (PMID 37225052). This information has now been included (lines 25-26 and 64-66).

In the materials and methods, please provide the exact time of treatment instead of “when pain like behavior was evident”

Response to Reviewer: The exact time of treatment has now been included (lines 107-108).

I suggest to change from “pain-like” to “pain-related” behavior.

Response to Reviewer: Thank you for this reasonable suggestion. However, we have used “pain-like” behaviour in our prior publications (see, for example, PMID 32028021 and PMID 37225052) and, for reasons of consistency, we prefer to continue using this term. Other research groups also use our terminology (see, for example, PMID 23169679). 

For statistical analyses, did the author test data distribution before applying non-parametric test?

Response to Reviewer: Yes, the test of data distribution was performed using a Shapiro-Wilk and Levene’s test. This information is now available in Material and Methods (lines 143-145). 

The authors have not included females in this study, which is a major limitation. There are reports which prove that female do not develop or do develop OA in DMM model.

Response to Reviewer: We agree that the exclusion of female mice is a major limitation – this exclusion is now mentioned in the Discussion (lines 239-243).

Results:

Line 118-124: This whole paragraph is based on data not shown. the CCL17 mAb mediated inhibition is important for this study. If the authors believe that this is important to be discussed in a separate paragraph under a subheading, the authors should include the data in the manuscript instead of saying data not shown.

Response to Reviewer: Many thanks for this comment. In response, we have now moved this paragraph into Materials and Methods as information about the purification and the inhibitory effects of B293 (see lines 72-87). 

Line 126, authors are proposing to study the effect of B293 on the development of OA in DMM model, but administer B293 after 9 week of DMM. Multiple studies have shown that mice already develop moderate to severe OA at 8-week post DMM.

Response to Reviewer: We agree that moderate to severe OA can be observed at 8 weeks post DMM. However, one of the aims of our study was to determine if CCL17 neutralization would affect both pain and disease in OA mice; therefore the chosen time point for anti-CCL17 mAb administration was based on when pain is robustly evident in our particular mice (i.e. 9 weeks post DMM surgery) to ensure that the therapeutic inhibitory effect on OA pain is clearly captured. 

Authors have not included any data on the severity of OA at 8-week post DMM to set a baseline for the study. This time point is important to understand if B293 is inhibiting further degradation of the cartilage or the protection is through some other mechanism.

Response to Reviewer: We agree that having data on the severity of OA at 8-week post DMM to set a baseline would be ideal to indicate a reduction in the further degradation of cartilage. However, our data showed that B293 treatment commencing at week 9 led to reduced cartilage degradation at termination, which we have now clarified (see lines 165 and 187). 

There is no sham control included in this study.

Response to Reviewer: We have shown previously that, in comparison to mice induced with DMM OA, mice with sham surgery showed no disease nor pain-like behaviour development (PMID 37225052). Given that the aim of this study was to focus on the treatment efficacy towards OA pain-like behaviour and disease, a sham control group was therefore not included again. However, we agree that ideally a sham control group should be included as a comparison. We have now included this information (see lines 159-160 and 184-185).

There is no no-treatment control included in this study.

Response to Reviewer: In this study, we used BM4 (25mg/kg) as a treatment control to assess the efficacy of different doses of B293 (see Figures 1 and 2). In previous publications (PMID 29622035 and PMID 32028021), we have found no difference in pain-like behaviour nor disease severity between PBS-treated and isotype control-treated groups; importantly, we observed differences between BM4-treated and B923-treated mice, indicating a role for CCL17. 

The whole joint images make it difficult to see if there is any damage to the cartilage in their DMM model. The difference in the safranin o staining intensity appears be due to overstaining of safranin o as can be seen by the strong staining of growth plate which can’t be seen in BM4 group image. Authors should provide better images to show cartilage damage, osteophytes.

Response to Reviewer: As requested, better histologic images have now been included in Figure 1B. We have also indicated cartilage damage and osteophytes with black and orange arrows, respectively (see Figure 1B). As indicated in the Supplemental Materials and Methods, the OARSI scoring system is mainly based on the intactness of cartilage surface with a minor component being the safranin O staining intensity.

This reviewer does not understand the rationale of combining DMM with HFD-induced OA. HFD-induced OA fits well with aging-related spontaneous OA.

Response to Reviewer: Our rationale for this study is to determine the efficacy of anti-CCL17 mAb therapy in an OA mouse model, namely the DMM-induced OA (see Figure 1). Given that obesity is known as a comorbidity for OA (PMID 34405518) and, based on our prior evidence implicating CCL17 in obesity-exacerbated DMM-induced OA (PMID, we therefore sought to assess also the efficacy of anti-CCL17 mAb therapy in this context with this information being now included (see lines 175-180) 

Same comments for images in figure 2.

Response to Reviewer: As requested, better histologic images have now been included in Figure 2. We have also indicated cartilage damage and osteophytes with black and orange arrows, respectively (see Figure 2B).

This reviewer does not understand why the results for “CCL17 and HFD-induced metabolic changes” were put in the supplementary figure.

Response to Reviewer: As recommended, Figures associated with “CCL17 and HFD-induced metabolic changes” have now become main Figures (see new Figures 3 and 4).

---

## [Editor Report · Decision Letter 1]

29 Dec 2024

Therapeutic blockade of CCL17 in obesity-exacerbated osteoarthritic pain and disease

PONE-D-24-36038R1

Dear Dr. Lee,

We’re pleased to inform you that your manuscript has been judged scientifically suitable for publication and will be formally accepted for publication once it meets all outstanding technical requirements.

Kind regards,

Xindie Zhou

Academic Editor

PLOS ONE
---

## [Editor Report · Acceptance letter]

7 Jan 2025

PONE-D-24-36038R1 

PLOS ONE

Dear Dr. Lee, 

I'm pleased to inform you that your manuscript has been deemed suitable for publication in PLOS ONE. Congratulations! Your manuscript is now being handed over to our production team.

Kind regards, 

on behalf of

Dr. Xindie Zhou 

Academic Editor

PLOS ONE